# Highly Efficient Anchor-Free Oriented Small Object Detection for Remote Sensing Images via Periodic Pseudo-Domain

Minghui Wang [1], Qingpeng Li [2,*], Yunchao Gu [1] and Junjun Pan [1]

1. The State Key Laboratory of Virtual Reality Technology and Systems, School of Computer Science and Engineering, Beihang University, Beijing 100191, China; minghuiw@buaa.edu.cn (M.W.); guyunchao@buaa.edu.cn (Y.G.); pan_junjun@buaa.edu.cn (J.P.)
2. The State Engineering Laboratory of Robot Vision for Perception and Control, School of Robotics, Hunan University, Changsha 410082, China
* Correspondence: liqingpeng@hnu.edu.cn

**Abstract:** With the continuous progress of remote sensing image object detection tasks in recent years, researchers in this field have gradually shifted the focus of their research from horizontal object detection to the study of object detection in arbitrary directions. It is worth noting that some properties are different from horizontal object detection during oriented object detection that researchers have yet to notice much. This article presents the design of a straightforward and efficient arbitrary-oriented detection system, leveraging the inherent properties of the orientation task, including the rotation angle and box aspect ratio. In the detection of low aspect ratio objects, the angle is of little importance to the orientation bounding box, and it is even difficult to define the angle information in extreme categories. Conversely, in the detection of objects with high aspect ratios, the angle information plays a crucial role and can have a decisive impact on the quality of the detection results. By exploiting the aspect ratio of different targets, this letter proposes a ratio-balanced angle loss that allows the model to make a better trade-off between low-aspect ratio objects and high-aspect ratio objects. The rotation angle of each oriented object, which we naturally embed into a two-dimensional Euclidean space for regression, thus avoids an overly redundant design and preserving the topological properties of the circular space. The performance of the UCAS-AOD, HRSC2016, and DLR-3K datasets show that the proposed model in this paper achieves a leading level in terms of both accuracy and speed.

**Keywords:** deep learning; remote sensing; arbitrary object detection; convolutional neural network

## 1. Introduction

Large Language Models (LLMs) such as ChatGPT and IndustrialGPT [1] exhibit notable performance in both language and vision tasks; however, their extensive memory consumption and computational burden impede their applicability in various tasks under limited computational resources. As a fundamental task in computer vision and remote sensing, object detection has achieved continuous progress. In recent years, researchers in this field have shifted attention from horizontal object detection to arbitrary object attention [2–4]. For some densely arranged rectangular objects, an arbitrary-oriented object detection method can match boundary better and distinguish those objects from each other.

To describe a rotated box, it is common practice to add the rotation angle parameter to the horizontal box [3] or to describe the coordinates of all four points [4]. Diverging from traditional bounding boxes, the description of arbitrary-oriented bounding boxes, commonly referred to as oriented bounding boxes (OBBs), poses significant challenges in the form of Periodicity of Angular (PoA) and EoE (Exchangeability of edges) problems. Furthermore, the existence of square or circular targets poses an additional challenge in that they frequently do not necessitate, and in some instances, cannot identify their orientation angle.

To address the aforementioned problems related to predicting oriented bounding boxes, various methods have been proposed. For instance, R2CNN [5] and oriented RCNN [6] employs redundant anchors alongside regression of orient box offsets, while CSL [7] tackles the problem by converting it from regression into a classification question. Guo et al. [8] use a convex hell to represent the oriented boxes. PIoU [9] computes a novel pixel-IoU loss for OBB targets. R3Det [4] proposes a rotation detector using a coarse-to-fine approach. GWD [10] and KLD [11] convert oriented boxes into 2D Gaussian Distribution and Kullback–Leibler Divergence, respectively. Moreover, CFC [12] directly computes the rotation angle loss utilizing the $\tan\theta$ function.

As shown in Figure 1, for targets with a high aspect ratio, the rotation angle is not only important but also accompanied by an evident visual feature. Nevertheless, two more points need to be made clear: first, the evident periodic is $\pi$ but not $2\pi$; second, for targets with a low aspect ratio, the rotation angle is not only unimportant, but the visual features are also relatively messy. Our work is motivated by these two points.

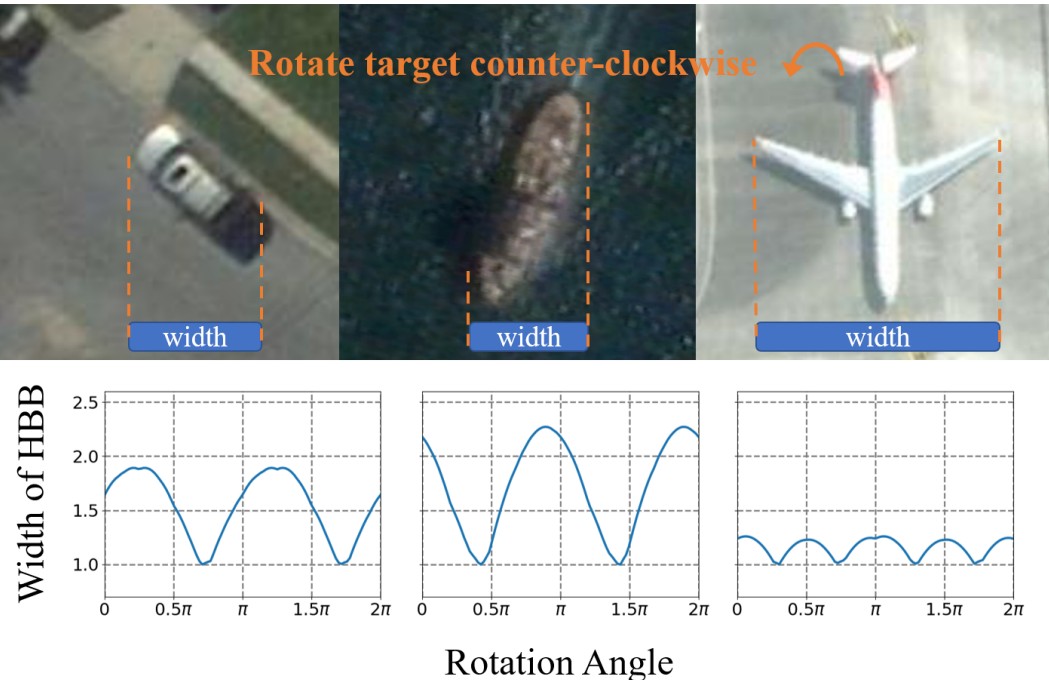

**Figure 1.** Three curves for width at different rotate angles(car, ship, airplane). The y coordinate denotes the width of the object bounding boxes, i.e., the length of the object's projection on the horizontal axis, where the minimum projection length is standardized to 1. For high aspect ratio targets, the width curve is closer to the trigonometric function. Otherwise, the width curve is more messy.

This paper proposes a topology-based detection method that utilizes a periodic pseudo-domain. In topology, it is well-known that the genus of the real number axis is zero, whereas the genus of the circular space is one. Consequently, using a single real number to regress the angular value accurately is not feasible. Instead, we utilize the natural embedding of the circular space in a two-dimensional Euclidean space to precisely estimate the angles of the oriented boxes. The weight assigned to the angle is determined according to the ratio of the height and width of the targets .

In the ensuing section, some work related to object detection and oriented object detection will be described. Later, a detailed explanation of the regressor's definition and loss function can be found. In Section 4, our method's performance and speed are evaluated through experiments on UCAS-AOD [13], HRSC2016 [14], and DLR-3K [15]. Section 5 provides a discussion of the benefits and drawbacks of the approach.

## 2. Related Work

As a fundamental task in computer vision, object detection has a wealth of research papers. In the following section, we will describe the representative research work on object detection and then some recent work on oriented object detection.

### 2.1. Object Detection Method

The goal of object detection is to find the location of the interested target in the image and use a bounding box to describe the pixel dimensions and coordinates of the target. As shown in Figure 2, two pairs of line segments parallel to the horizontal and vertical axes of the image are used to form the minimum bounding boxes of the target. The most mainstream methods of current object detection are the two-stage and one-stage methods. We will describe the advantages and disadvantages of these methods and their representative work, respectively.

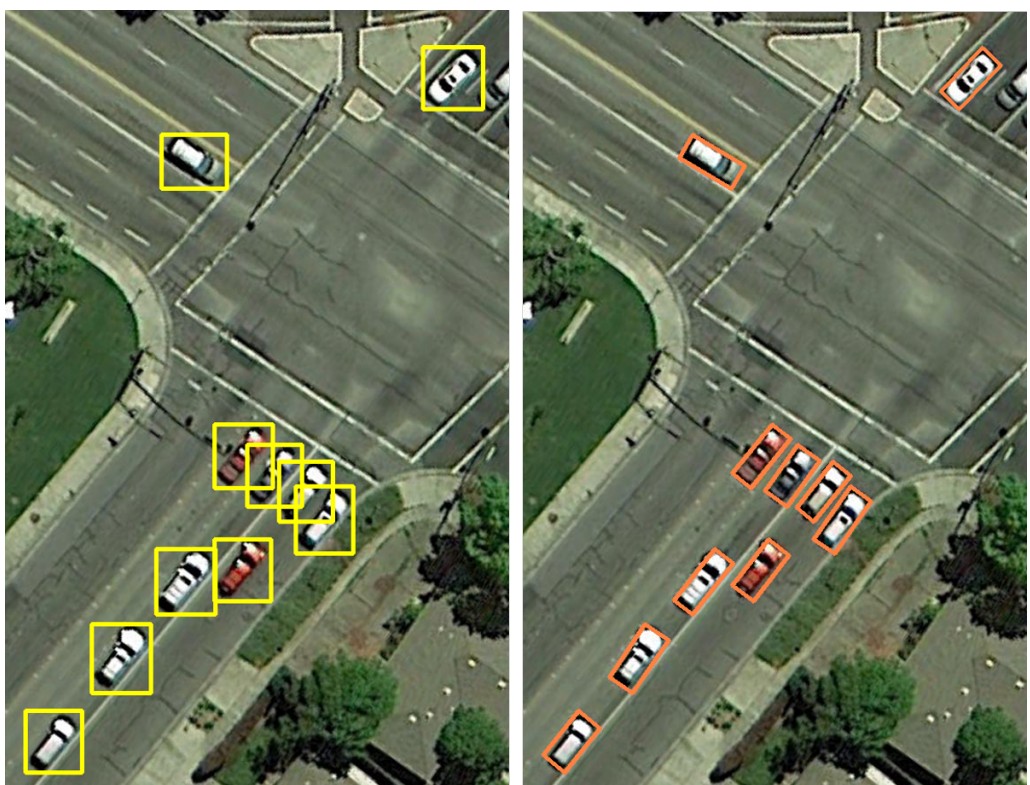

**Figure 2.** (**Left**) HBB: horizontal bounding boxes. (**Right**) OBB: oriented bounding boxes.

The two-stage work uses a convolutional network named Region Proposal Network (RPN) to propose candidate regions in the first stage, after which each proposal is evaluated and refined in the second stage to obtain the final detection results. The most well-known two-stage models include R-CNN [16], fast-RCNN [17], faster-RCNN [18] and R-FCN [19]. Faster RCNN achieved state-of-the-art results on the PASCAL VOC dataset [20] during that period and became a standard for two-stage models. Thousands of proposals need to be extracted in the first stage, thus ensuring accuracy while making the two-stage methods more time-consuming.

The most famous one-stage methods are YOLO [21], YOLO9000 [22], YOLOv3 [23], SSD [24] and RetinaNet [25]. The one-stage methods are more straightforward: they divide the image into grids and predict the target's confidence, center offset, width, and height within each grid. The one-stage methods infer much faster than the two-stage methods but also have a relatively lower accuracy than the two-stage methods for targets with significant intra-class variations.

In addition to the proposal-based two-stage methods and the grid-based one-stage methods, researchers have proposed many other approaches. For instance, CornerNet [26]

estimates the bounding box of an object by predicting a pair of points in the upper left and lower right corners of the target. Furthermore, CenterNet [27] uses a triplet of the object centroid and a pair of corner points to predict the bounding box of an object. ExtremeNet [28] uses five key points to estimate the bounding box.

### 2.2. Oriented Object Detection

The purpose of oriented object detection is also to find the location of the interested target and label it. However, the difference is that the two pairs of line segments are no longer parallel to the horizontal and vertical axes of the image. As shown in Figure 2, oriented bounding boxes may fit objects closer. Even in dense scenes, oriented bounding boxes rarely overlap with those of surrounding targets. Oriented object detection is generally used most in scene text [2,5,29], and aerial images [3,4,12].

The current representation of oriented object detection is less uniform than horizontal object detection. This is mainly due to the periodicity of angles. For ease of understanding, we will use the 360-degree representation of an angle as an example: when the angle is increased by 5 degrees from 358 degrees, the object is rotated by only a slight angle, but the value of the angle suddenly changes from 358 to 3 degrees. Since the deep convolutional network is, in fact, a continuous mapping, according to the intermediate value theorem for continuous functions, during a slight change in target rotation, the value of the angle also continuously passes through all real values between 3 and 358 degrees. This is far from what we would expect from the loss function, and thus often results in significant loss increases. Further, by topology, we know that the unit circle is not homeomorphism with any subset of the real numbered axes. This means that any direct regression of the angle using a single real number will inevitably lead to a significant loss increase under the boundary conditions. In order to solve the above oriented boxes representation challenges, researchers have taken various approaches, including but not limited to the following: dense proposal methods, Gaussian-like heatmap methods, point-based representation methods, regression-to-classification methods(CSL), and polar coordinate representation methods. This subsection reviews these commonly used methods for oriented object detection.

Influenced by faster RCNN [18], oriented object detection has many two-stage methods. For example, oriented RCNN [6] is a typical one, which includes an oriented Region Proposal Network (RPN) in the first stage. Pan et al. [30] develop a dynamic refine network for dense objects. The advantages and disadvantages of the two-stage method in oriented object detection are similar to the two-stage way in a horizontal one. Guo et al. [31] develop a dense proposal method for vehicle detection based on orientation-aware feature fusion. Wang et al. [32] also use a two-stage method for small and dense building detection via horizontal bounding boxes.

In addition to the oriented proposal, many point-based arbitrary orient detection approaches exist. Li et al. [33] represents the original oriented bounding boxes by dynamically assigning multiple points to the boundaries. Zand et al. [34] and APS-Net [35] utilize five and nine points to represent an oriented bounding box separately. Guo et al. [8] tried to find a convex hull representation for the oriented bounding boxes. Refs. [36,37] are also point-based work.

Another common idea is to develop the loss function for arbitrary-oriented object detection. For example, ref. [38] use smooth L1 loss; PIoU [9] use Pixels-IoU loss; RAIH-Det [39] use cyclical focal loss; and Refs. [40,41] use GIoU loss. Shi et al. [42] employ smooth L1 loss to arbitrary oriented vehicle detection. Zhang et al. [43] propose an oriented infrared vehicle detection method. CSL [7] predicts angle via classification instead of regression. Chen et al. [44] apply CSL on ship detection task.

There are also a lot of Gaussian-like methods, such as [10,45–47], who use Gaussian Heatmap for the arbitrary-oriented object detection task. KLD [11] represent oriented boxes as Kullback–Leibler Divergence. Guo et al. [48] improve an orientation-aware Gaussian heatmap (OAG) method for vehicle detection task.

One interesting thing about OBBs is that Cheng et al. [49] found that oriented bounding boxes (OBBs) only appear diagonally from the horizontal bounding boxes (HBBs) if these two do not coincide. Awareness of this fact can help one to propose proposals more efficiently. Nie et al. [50] use two HBBs to represent an OBB. Li et al. [51] utilize background information to assist in the detection. Pu et al. [52] enhance adaptive rotated convolution for rotated object detection. Yao et al. [53] use two symmetrical offsets in a polar coordinate system to represent the OBB.

## 3. Materials and Methods

As we explained in the previous subsection, the prediction of angular information cannot be made by directly regressing a single real number. So it is natural to think, what is the minimum number of real numbers that can be regressed to achieve stable prediction of angular information? In this paper, we propose a method to predict the rotation angle of an oriented bounding box by two real numbers. The space formed by these two real numbers is mathematically known as the real projective line (RPL), which will be called the periodic pseudo-domain in this paper. The domain is similar to the unit circle but has a period of $\pi$ instead of $2\pi$. The main reason for using $\pi$ instead of $2\pi$ in our method, as shown in Figure 1, is that for the visual features of the oriented bounding box, making the period in $\pi$ coincides with the period of the visual features and thus does not cause significant loss increases.

Our proposed approach for oriented object detection in aerial imagery applies an anchor-free trainable network. As shown in Figure 3, we introduce the end-to-end method framework in Section 3.1. The details in our detection head and loss functions are explained in Sections 3.2 and 3.3, separately.

### 3.1. Framework

The main process of our method is shown in Figure 3. There are a total of 5 steps between input images and output-oriented boxes which were displayed column by column. In pre-process and post-process steps, there have no trainable parameters. All deep neural network computations are calculated during the second step and fourth steps.

In the pre-processing step, we process the image to make sure it can be fed into GPU correctly. First, if the image size is larger than the setting value, we resize the image and keep the aspect ratio. Then we padded zeros at the right and bottom edge of the scaled image to make sure both image height and width can be divided by 32.

Our method use a pre-trained deep neural network as backbone, which may include but is not limited to resnet18, reset50, resnet101, and darknet53 [23,54]. The backbone may largely determine the model speed, performance, convergence time, and others.

We pass the visual feature at 1/8, 1/16, and 1/32 resolutions from backbone to feature pyramid network (FPN) neck [55]. Feature vector at each position of those resolutions represents a grid that has $8 \times 8$, $16 \times 16$, and $32 \times 32$ pixels, separately. We unify the dimension of the above feature and concatenate different level features before feeding them into heads.

In CR head step, we use both contour and rectangle heads at each FPN level. We classify the foreground and background for each category at the contour head, and regress the center offset, breadth, length and rotate angle of the rectangle at the other head. The proposed CR head will be introduced in detail in the next section.

After model inference, we concat all prediction rectangle together. Then we use confidence threshold and skew-IoU [2] based non-max-suppression to the predict boxes. It is worth mentioning that calculating for skew-IoU is much more time-consuming than horizontal IoU, such that we also calculate it with the GPU.

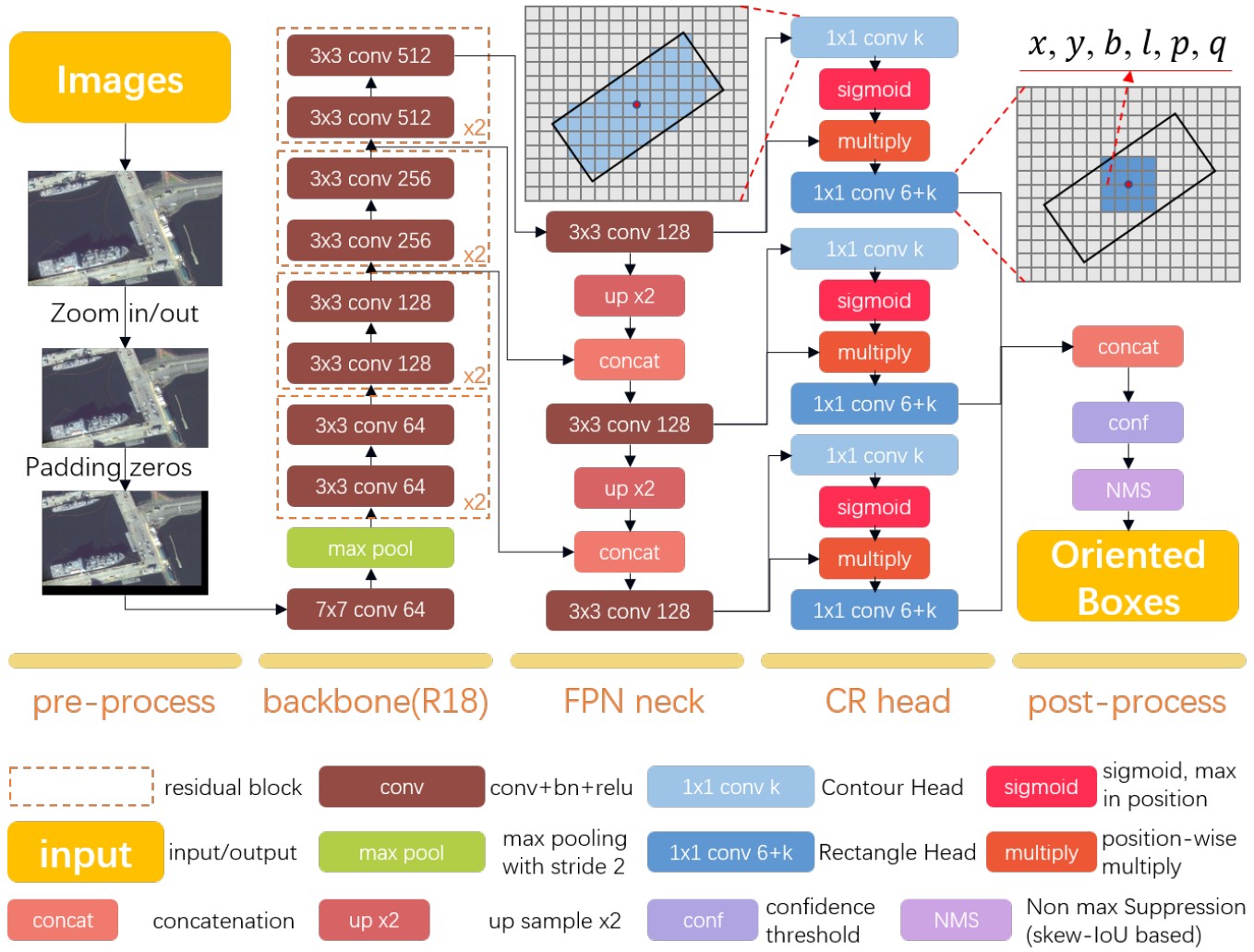

**Figure 3.** Framework of the proposed method with backbone resnet18. $k$ means the number of categories. At the Contour and Rectangle (CR) head, the confidence for each category is computed separately. $x,y$ denote the rectangle center offset at $x$ and $y$ axis. $b$, $l$ denote the breadth and length of the rectangle. $p$, $q$ are the regressors of $\cos 2\theta$ and $\sin 2\theta$, where $\theta$ represents the rotate angle between the long side and $x$ axis.

### 3.2. CR Head and Pseudo-Domain

Two issues need to be addressed to design an arbitrary direction object detector, i.e., periodicity and degeneration.

The first issue is how to express and predict the angle. As we all know, the degree of the angle has its periodic, but the real number axis has an ordered structure. That means the ring space and 1-dim linear space are topologically different. Thus, the angle of the oriented object bounding box cannot be determined by predicting a single real number. To address this issue, we use the natural embedding of the circular space in a two-dimensional Euclidean space to regress the degree of the angle. It is noteworthy that the periodic in rotate rectangle is $\pi$ instead of $2\pi$. More specifically, for a given angle $\theta$, we predict $\cos 2\theta$ and $\sin 2\theta$ in a pseudo-domain.

Furthermore, the other issue is that different kinds of remote-sensing object have different aspect ratios. For example, the aspect ratio of objects such as cars and ships may reach 2:1, 3:1, or even higher, while the aspect ratio of objects such as airplanes, storage tanks, roundabout may be very close to 1:1, and sometimes the the angle of the target may not even be defined. In the former, the angle of the oriented bounding box is essential and distinctive, while in the latter, the angle of the bounding box is irrelevant and

weakly distinctive. In the next subsection, we will propose an angle loss to address the degeneration issue and make a trade-off between those two kinds of objects.

To better encode the rotational rectangle information, we use both the contour head and the rectangle head. For $k$ given category, we predict $k$ float numbers in each grid at contour head and $k + 6$ float numbers in each grid at the rectangle head.

The contour head is a binary classification approach, each category has its own heatmap. The target is set to be 1 if the grid center locates in a rectangle, otherwise 0.

The rectangle head predicts $k + 6$ float number in each grid, where the first $k$ float numbers denote the confidence of the rectangle for each category, separately. The 6 extra float number $(x, y, b, l, \cos 2\theta, \sin 2\theta)$ in this head represents the rectangle shape, where $x, y$ predict offset at $x$ and $y$ axis for rectangle center, $b, l$ predict the breadth and length of the rectangle, and the last two float numbers represent the rotation angle of the rectangle.

### 3.3. Loss Function

Our arbitrary-orientation method has 3 contour heads and 3 rectangle heads. We compute loss at each head and add them together to obtain the joint loss.

At both rectangle and contour heads, we use the cross-entropy loss for each category, separately. Furthermore, also, this is the only loss computed by contour heads.

$$L_{conf} = -\sum_{k=1}^{K} \left( \frac{\sum_{i \in G_k^+} \log s_k^i}{|G_k^+|} + \lambda \frac{\sum_{i \in G_k^*} \log (1 - s_k^i)}{|G_k^*|} \right) \tag{1}$$

where $G_k^+$ is the set of all grids whose target is set to be 1 in category $k$, $G_k^*$ is the set of all grids whose target is set to be 0 in category $k$, $K$ is category numbers, $s_k^i$ is the grid confidence for category $k$ in grid $i$, and $\lambda$ is a given constant.

At rectangle heads, we regress the angle in a pseudo-domain to address its periodicity and degeneration issues.

$$L_{ang} = \frac{1}{|G^+|} \sum_{i \in G^+} \left[ \log(\frac{l_i^*}{b_i^*}) \right]^\gamma (|p_i - \cos 2\theta_i^*| + |q_i - \sin 2\theta_i^*|) \tag{2}$$

where $p_i, q_i$ is the predict result, $b_i^*, l_i^*$ denote the breadth and length of the rectangle, $\theta_i$ represents the rotate angle between the long side and x axis, and $\gamma$ is a given constant.

As shown in Figure 4, the angle has more loss when the aspect ratio is larger. In response to this, for circular and square objects, angle loss has less weight. We use a penalty loss, as shown below, to constrain the parameters $(p, q)$ within a periodic pseudo-domain rather than the entire two-dimensional Euclidean space:

$$L_{pen} = \frac{1}{|G^+|} \sum_{i \in G^+} \log(\frac{l_i^*}{b_i^*})(|p_i^2 + q_i^2 - 1|) \tag{3}$$

The regression loss for offset parameters at each rectangle head is computed in each grid separately as below

$$L_{off} = \frac{1}{|G^+|} \sum_{i \in G^+} \sum_{t \in \{x_i, y_i\}} (t - t^*)^2$$
$$+ \frac{1}{|G^+|} \sum_{i \in G^+} \sum_{t \in \{b_i, l_i\}} (\log t - \log t^*)^2 \tag{4}$$

where $G^+$ is the set of all grids whose target is set to be 1, $x_i^*, y_i^*$ is the ground truth for rectangle center offset at x-axis and y-axis in grid $i$, $b_i^*, l_i^*$ is the ground truth for breadth and length for rectangle in grid $i$, and $x_i, y_i, b_i, l_i$ is the corresponding predict result in grid $i$.

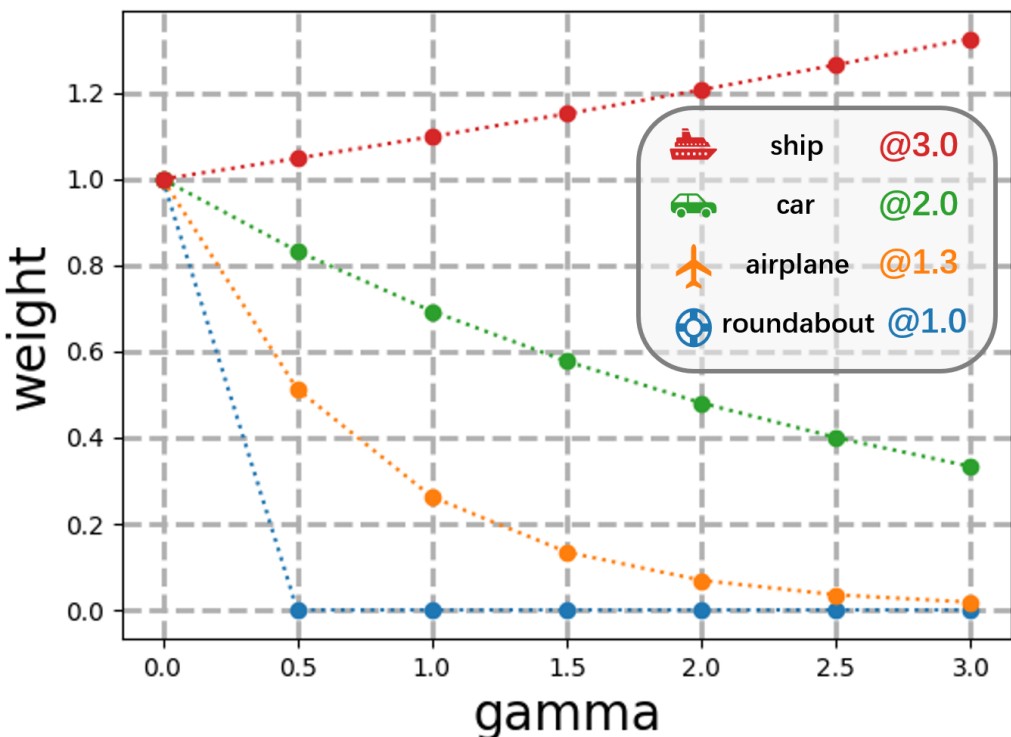

**Figure 4.** The penalty loss to the pseudo-domain is to apply a smaller weight for targets with low aspect ratios, like roundabouts and airplanes, and a larger weight for targets with high aspect ratios, such as cars and ships.

The final joint loss is computed as below:

$$Loss = \sum_{h \in \mathbb{C}} L_{conf}^h + \sum_{h \in \mathbb{R}} \left( L_{conf}^h + L_{ang}^h + L_{pen}^h + L_{off}^h \right) \tag{5}$$

where $\mathbb{C}$ represents the set of contour heads and $\mathbb{R}$ represents the set of rectangle heads.
Figure 5 shows the training loss with different backbones.

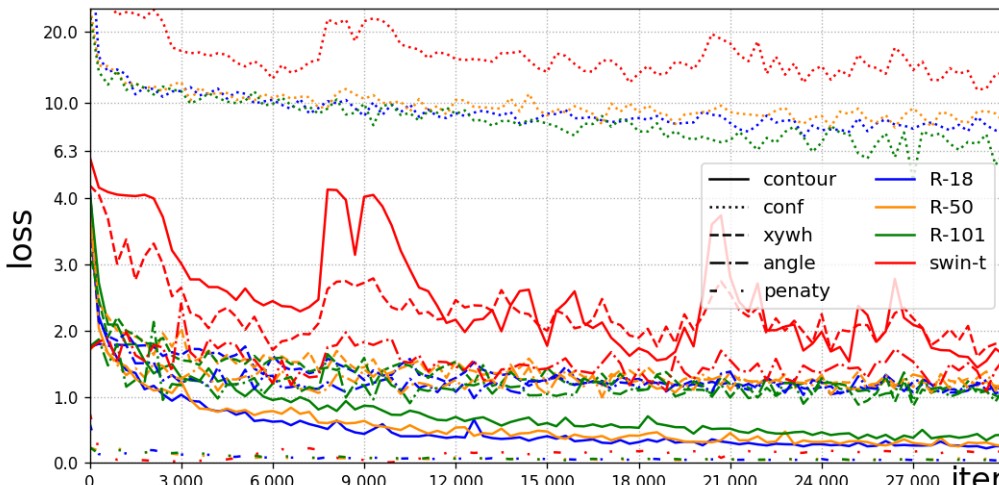

**Figure 5.** All parts of training loss with different backbones. The shown loss was averaged for every 300 iters. We perform a logarithmic treatment for loss greater than 4 in this figure.

## 4. Results

All experiments were implemented on a single Titan X with 12 GB GPU memory, Intel(R) Core(TM) i5-7500 CPU @ 3.40 GHz with 16 GB CPU memory. We write the code under the software environment Python 3.9.0 and torch 2.0.0. All image scaling maintains the aspect ratio. We use graying, rotation, crop, and multiscale for data augmentation.

### 4.1. Dataset and Metrics

We evaluate our model performance on three public remote sensing datasets: UCAS-AOD, HRSC2016, and DLR-3K (Figure 6). The object in the above datasets was labeled by oriented bounding boxes.

The UCAS-AOD [13] dataset has 1000 plane images and 510 vehicle images. We randomly chose 755 (50%) images for training, 302 (20%) images for validation, and 453 (30%) images for testing.

The HRSC2016 [14] dataset is a challenging ship detection dataset, which provides 436, 181, and 444 images for training, validation, and testing separately.

The DLR-3K dataset [15] is a set of UAV images captured by a 3K+ camera system that has 10 raw images of size $5616 \times 3744$ and labels. We adopt five images and the corresponding labels for training and the other five for the test.

We use the VOC 2012 mAP metrics to evaluate the model performance, and compute IoU the same way as SkewIoU [2]. To evaluate the model inference speed, we use the key ms/Mpx to denote milliseconds per million pixels.

### 4.2. Ablation Study

We tested different $\gamma$ values for the pseudo adaptive on the UCAS-AOD dataset and trained all models using Adam for 300 epochs. The results, shown in Table 1, indicate that using the pseudo-domain method has the potential to help models balance both high and low aspect ratio targets. The performance of backbone swin-t [56] is weak, thus demonstrating that the swin transformer backbone does not work well in small datasets such as UCAS-AOD. It is interesting to note that through the ablation study, we have discovered that redundant network structures may negatively affect model performance when the dataset size is limited.

**Table 1.** The ablation results of our methods at different settings on UCAS-AOD dataset. The flops are computed with input shape $(1, 3, 512, 512)$.

| Backbone | $\gamma$ | Car | Plane | mAP | ms/Mpx | Params (M) | Flops (G) |
|---|---|---|---|---|---|---|---|
| | 1.0 | 84.48% | **97.44%** | 90.96% | | | |
| resnet18 | 2.0 | 86.41% | 95.60% | 91.00% | **16.21** | **12.5** | **22.7** |
| | 3.0 | **89.24%** | 96.76% | **93.00%** | | | |
| resnet50 | 1.0 | 83.84% | 96.61% | 90.22% | 63.16 | 44.8 | 101.2 |
| | 3.0 | 81.96% | 93.03% | 87.50% | | | |
| resnet101 | 2.0 | 77.29% | 92.97% | 85.13% | 83.54 | 63.8 | 140.2 |
| | 3.0 | 71.14% | 90.95% | 81.04% | | | |
| swin-t | 3.0 | 21.56% | 57.81% | 39.69% | 69.23 | 21.8 | 39.3 |

### 4.3. Performance

We assessed the mAP of our models on the three datasets mentioned above. The inference speed of our methods, as well as that of the compared methods, was tested on a single Titan X GPU. The detection results are shown in Figure 10. All categories of mAP in compared methods are reported in their own paper.

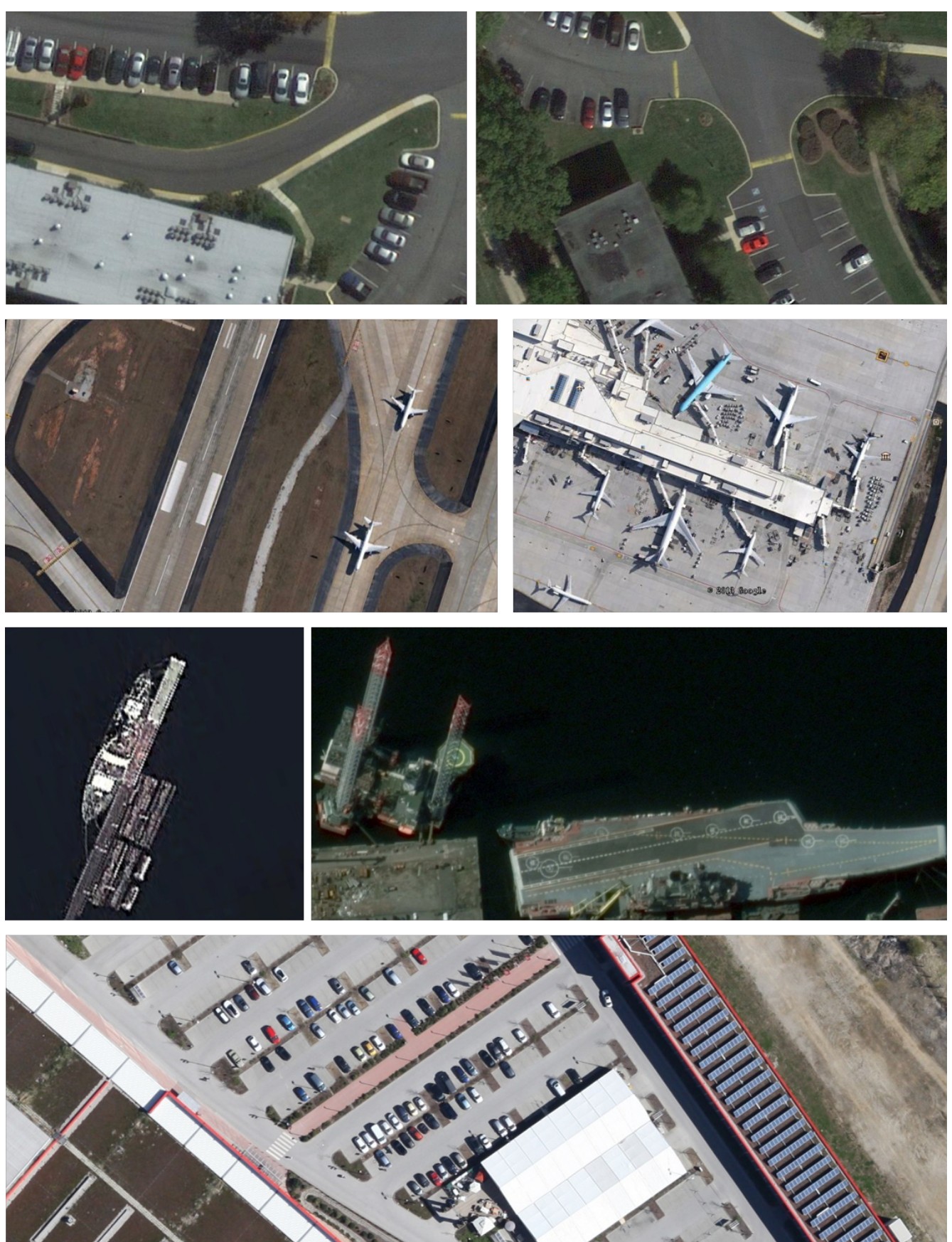

**Figure 6.** Example of dataset images. **Top 2 rows**: UCAS-AOD dataset; **third row**: HRSC2016 dataset; **last row**: DLR-3K dataset. The images are cropped and zoomed.

**Result from UCAS-AOD.** Our models were trained using Adam for 300 epochs, while $\gamma$ was set to be 3.0. The results in Table 2 and Figure 7 demonstrate that our method is competitive in both speed and performance. The test error bar is shown in Figure 8, and the precision-recall curves for the airplane and car categories can be observed in Figure 9.

**Table 2.** Performance on UCAS-AOD dataset. Key: ms/Mpx = millisecond per million pixels.

| Method | Backbone | Car | Plane | mAP | ms/Mpx |
|---|---|---|---|---|---|
| CFC-Net [12] | resnet50 | **89.29%** | 88.69% | 89.49% | 135.52 |
| DAL [57] | resnet50 | 89.25% | 90.49% | 89.87% | 113.97 |
| SLA [58] | resnet50 | 88.57% | 90.30% | 89.44% | 72.94 |
| ours | resnet18 | 89.24% | **96.76%** | **93.00%** | **16.21** |
| | resnet50 | 81.96% | 93.03% | 87.50% | 63.16 |
| | resnet101 | 71.14% | 90.95% | 81.04% | 83.54 |
| | swin-t | 21.56% | 57.81% | 39.69% | 69.23 |

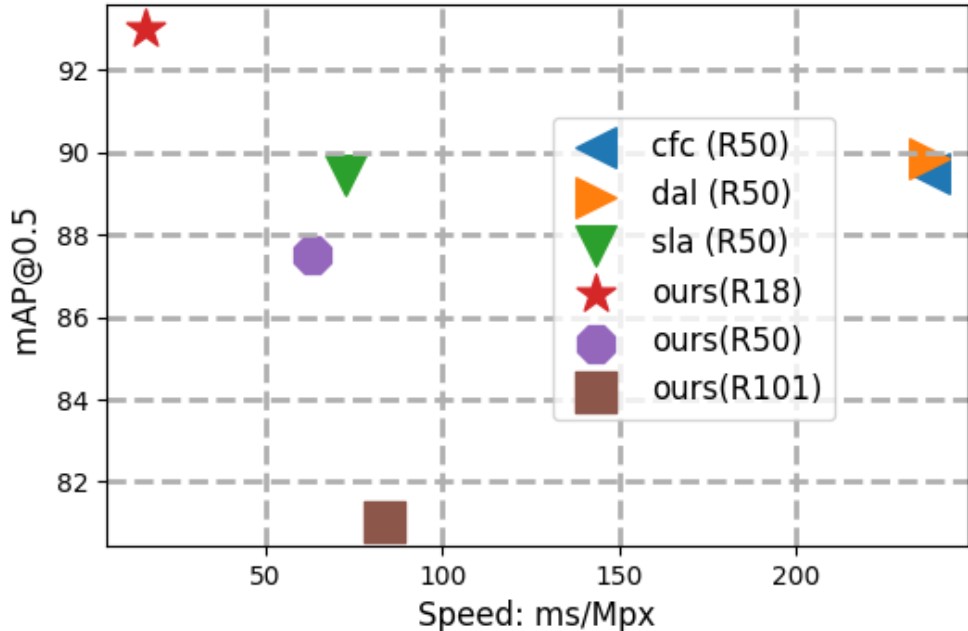

**Figure 7.** Results of the UCAS-AOD dataset, speed test on a single Titan X.

**Result from HRSC2016.** The result from HRSC2016 is reported in Table 3. The 181 images in the val part are not used for training. The results reveal that the accuracy of our method is relatively poor when dealing with targets that have high height-width-ratio targets but also have a good processing speed.

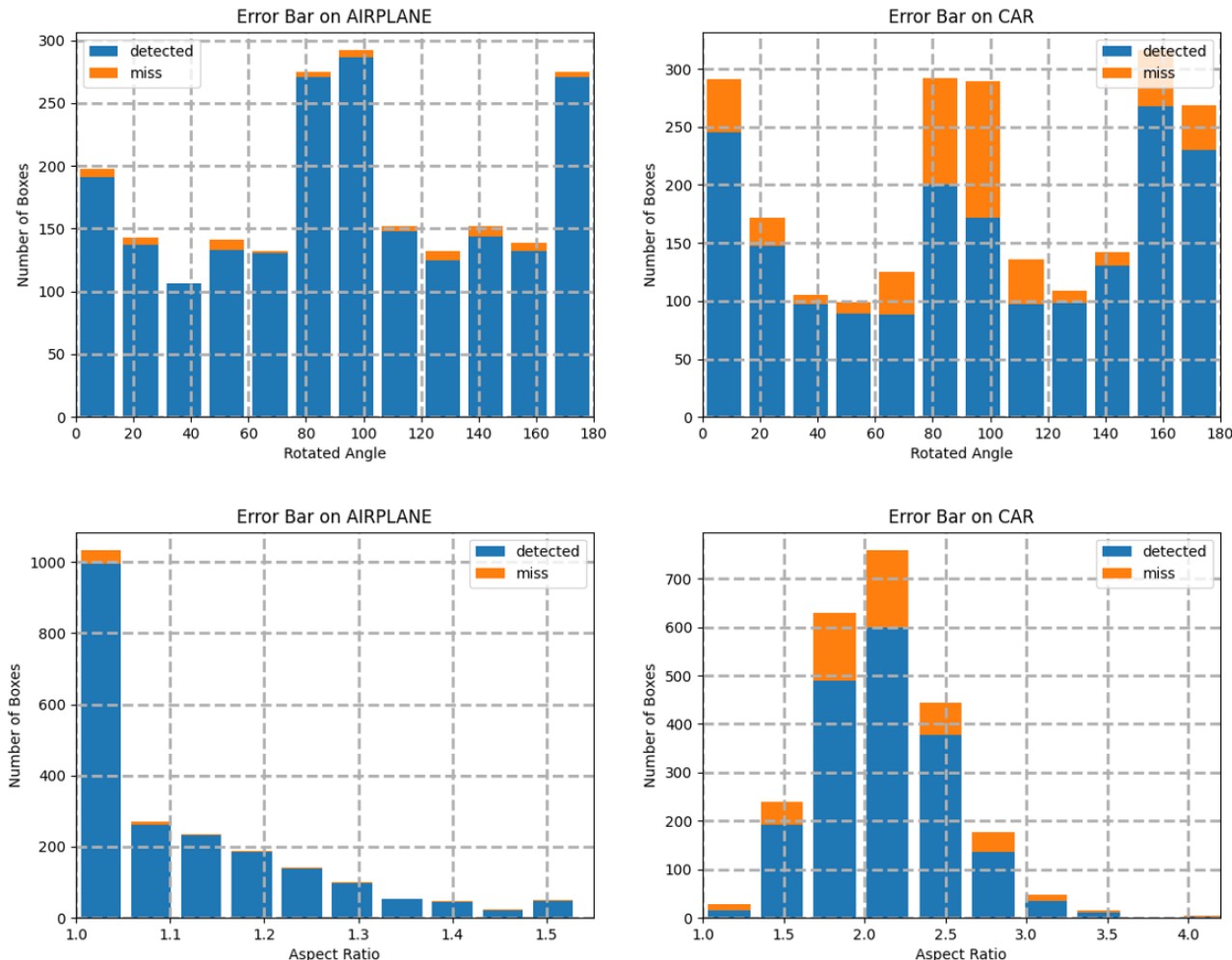

**Figure 8.** The test error bar on the UCAS-AOD dataset with backbone resnet18 ($\gamma = 3.0$).

**Table 3.** Performance on HRSC2016 and DLK 3K dataset.

| Dataset | Method | Backbone | mAP | ms/Mpx |
|---|---|---|---|---|
| HRSC2016 | CFC-Net [12] | resnet50 | 88.6% | 148.78 |
| | DAL [57] | resnet50 | 88.60% | 118.75 |
| | SLA [58] | resnet50 | 87.14% | 838.47 |
| | MRDet [59] | resnet101 | **89.94%** | 1249.36 |
| | ours | resnet18 | 78.07% | **17.53** |
| | | resnet50 | 73.59% | 65.87 |
| | | darknet | 61.15% | 61.31 |
| DLR-3K | ours | resnet18 | **83.99%** | **13.64** |
| | | resnet50 | 69.79% | 62.26 |
| | | resnet101 | 71.99% | 80.59 |
| | | darknet | 79.04% | 59.46 |

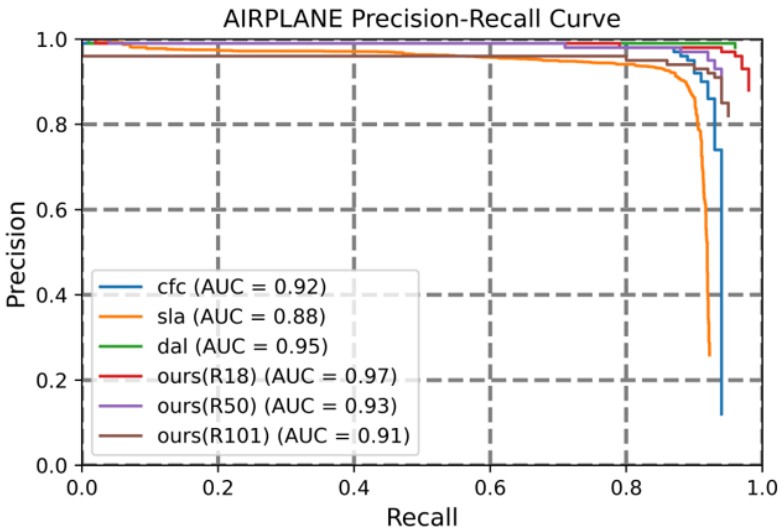

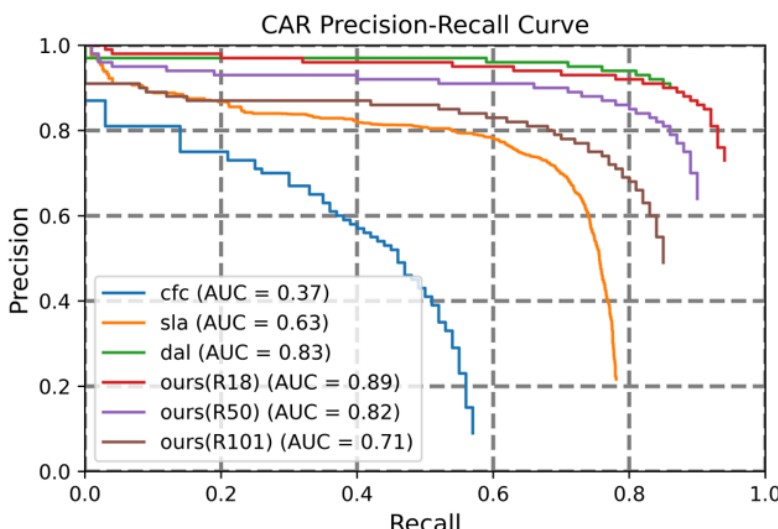

**Figure 9.** Results on the UCAS-AOD dataset, compared with our methods.

**Result from DLR-3K.** We evaluate our model on the small and dense arranged oriented detection dataset DLR-3K; the result is shown in Table 3 and Figure 10.

Furthermore, we also test the above UCAS-AOD dataset-trained model on aerial video frames of busy parking lot surveillance (https://www.youtube.com/watch?v=yojapmOkIfg (accessed on 29 June 2023)). The results are shown in Figures 11 and 12 and Table 4.

**Table 4.** Performance on video frame.

|         | TP  | FP | TN | Accuracy | Recall | F1 Score |
|---------|-----|----|----|----------|--------|----------|
| frame-A | 482 | 17 | 8  | 0.9659   | 0.9837 | 0.9747   |
| frame-B | 491 | 16 | 7  | 0.9684   | 0.9859 | 0.9771   |
| frame-C | 481 | 17 | 9  | 0.9659   | 0.9816 | 0.9737   |
| frame-D | 487 | 16 | 10 | 0.9682   | 0.9799 | 0.9740   |
| frame-E | 479 | 15 | 12 | 0.9696   | 0.9756 | 0.9726   |
| frame-F | 477 | 17 | 14 | 0.9656   | 0.9715 | 0.9686   |

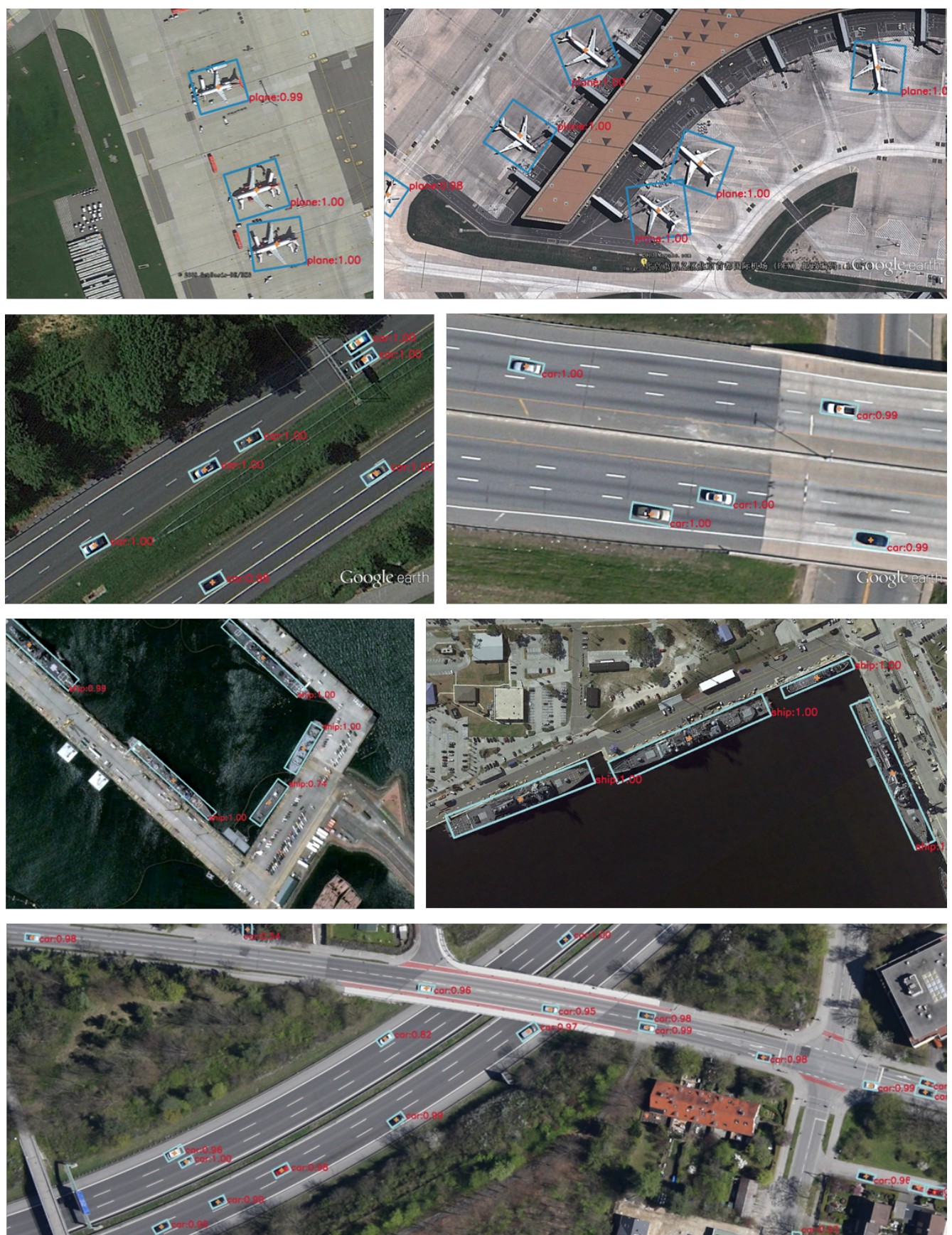

**Figure 10.** Detection results. **Top 2 rows**: UCAS-AOD dataset; **third row**: HRSC2016 dataset; **last row**: DLR-3K dataset. The images are cropped and zoomed.

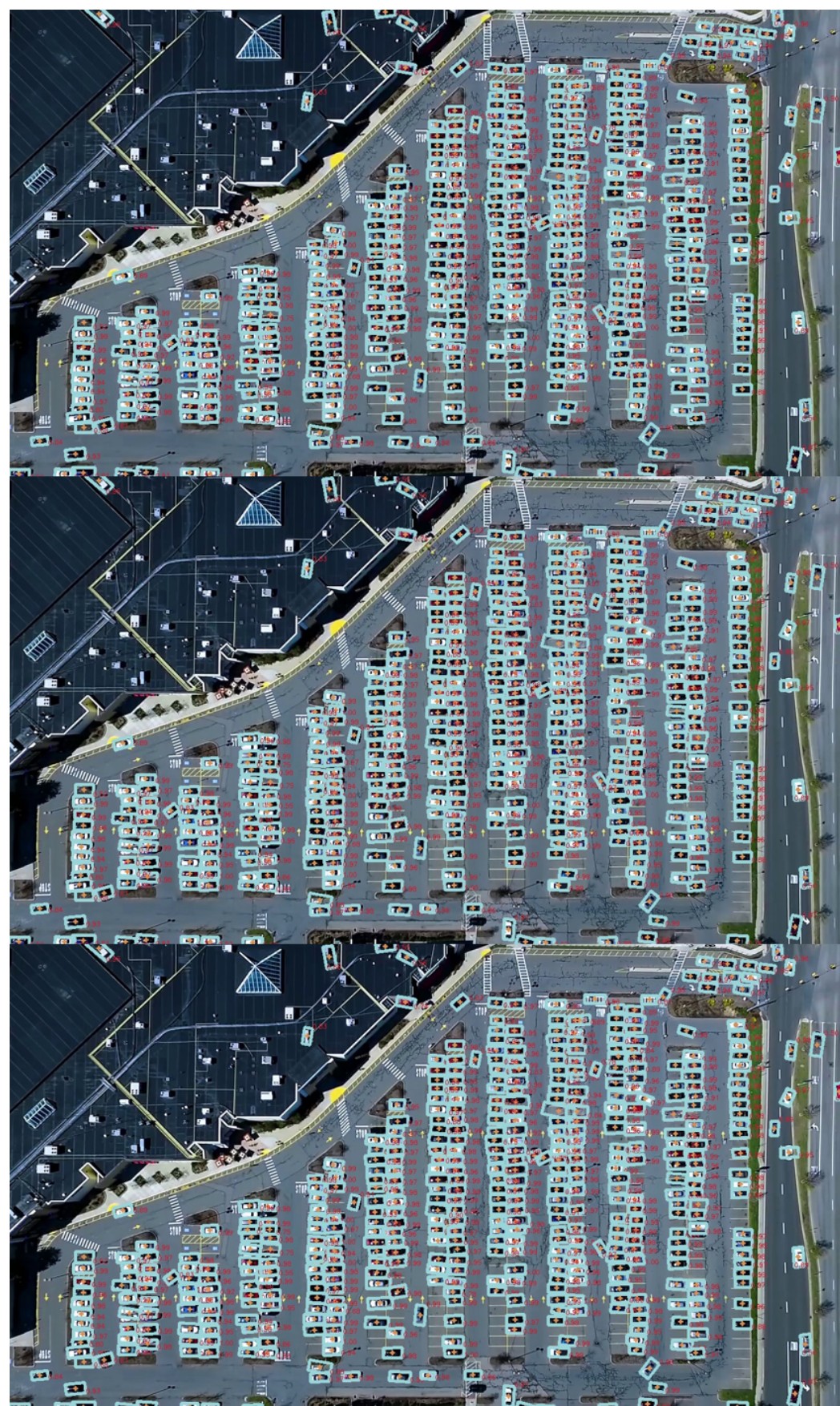

**Figure 11.** Detection result from aerial video frames without training. **Top**: frame-A; **second**: frame-B; **bottom**: frame-C.

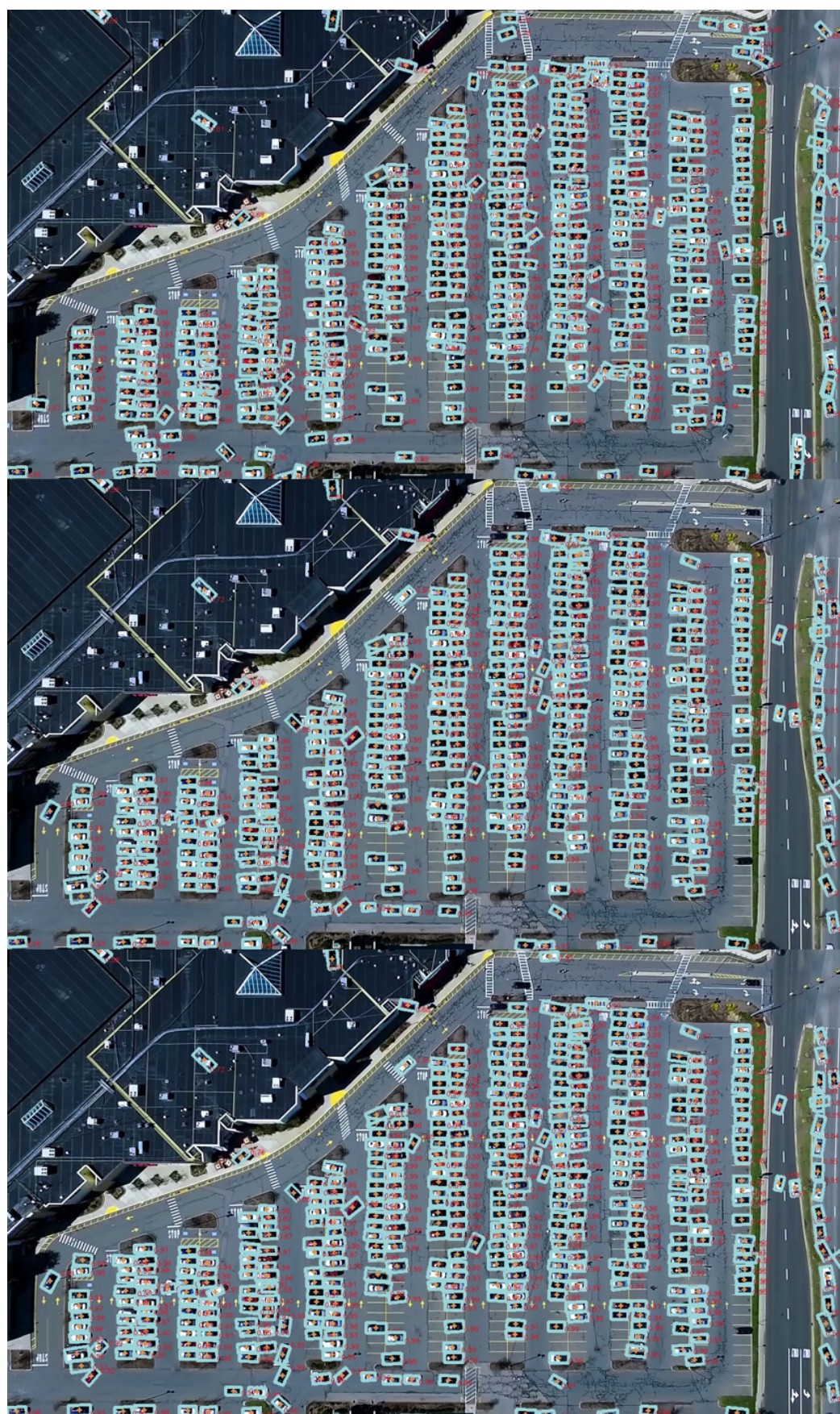

**Figure 12.** Detection result from aerial video frames without training. **Top**: frame-D, **second**: frame-E; **bottom**: frame-F.

## 5. Discussion

Results from several datasets have demonstrated that regression of angular information using two real numbers requires the fewest regression parameters theoretically and converges stably to the expected result. We think that there are two possible reasons for the effectiveness of the periodic pseudo-domain:

(a) The used periodic pseudo-domain is topologically homogeneous with the unit circle, thus avoiding the significant loss increase in boundary conditions;

(b) The setting of the period in the pseudo-domain is consistent with the period of the visual features of the target during rotation, which enhances the discriminative of the model during the angular regression process.

On the other hand, the setting of the regression parameters is kept as simple as possible in the design, thus avoiding additional redundant computations, leading to a faster processing speed of our algorithm.

One of the primary characteristics of the proposed technique is its speed. As shown in Figure 7, the straightforward architecture enables our method to surpass other methods in terms of speed. Another contradictory outcome, as demonstrated in Table 1, suggests that an increase in the complexity of the model backbone may lead to a decrease in the model's performance and speed. This phenomenon can be explained by the restricted size of the dataset and the use of a simple architecture in the detection method.

Another crucial attribute of the method is its ability to reconcile the significance of angular information for targets possessing varying aspect ratios. As an illustration, the UCAS-AOD dataset consists of two categories: cars and airplanes. The former classifies as a high aspect ratio target, whereas the latter does not possess such a characteristic. It follows that for the former category, the angular information of the oriented bounding box plays a significant role, while for the latter category, it is inconsequential. As shown in Table 2, our method balances these two categories favorably.

Improving horizontal object detection to oriented object detection is a practical task for remote sensing object detection. However, the angle parameter has more characteristics than the $xywh$ parameters. First is the periodicity of the angle itself. Secondly, the importance of angle information varies greatly for targets with different aspect ratios. We design a strategy to achieve a better and faster regression computation for the angle variables in the oriented bounding box. The design also provides a possible way for the subsequent researchers of oriented object detection to deal with such periodic variables.

In addition to this, one of the disadvantages of our approach is that its efficacy is comparatively lower for larger targets than for smaller ones. As shown in Table 3, although our method is also fast in handling large targets, the accuracy will be somewhat lower than the comparison methods. This suggests that in the case of oriented object detection for larger targets, albeit redundant candidates may consume additional time, they can offer a relatively favorable performance boost at this phase.

## 6. Conclusions

In this paper, we proposed a simple mathematical method to detect arbitrary direction objects in aerial images using a deep neural network. For more details, we embedded the rotation angle of each oriented object into a two-dimensional Euclidean space and regressed them with the deep network. That method not only preserves the typological properties of the circular space but also avoids an overly redundant design. Furthermore, we notice that the importance of the rotation angle in high-aspect and low-aspect ratio objects are different, thus we made a trade-off between those two kinds of objects. The detection results of the neural network are fed into a skew-IoU based NMS method to obtain the final result. The experiment on several remote sensing objects shows our arbitrary-oriented detection method makes a good performance in both speed and precision.

The experimental results demonstrate that it is possible to implement the regression computation of oriented object detection in a smaller design without adding too many redundant computations on top of the horizontal object detection. The research content of

this paper also provides a possible idea for subsequent researchers to design faster and more stable oriented object detection algorithms. With more lightweight models and compression algorithms, it can be developed into an approach more suitable for deployment on edge devices such as UAVs. In addition, a better trade-off between high and low aspect ratio targets will be an important research point in oriented object detection. In the future, we will look for suitable machine learning algorithms to improve the algorithm's ability to trade-off between multiple classes of targets more effectively.

**Author Contributions:** Conceptualization, M.W., Q.L. and Y.G.; methodology, M.W., Q.L. and J.P.; software, M.W.; validation, Q.L. and J.P.; formal analysis, M.W.; investigation, M.W.; resources, Q.L.; data curation, Y.G.; writing—original draft preparation, M.W.; writing—review and editing, Q.L.; visualization, M.W.; supervision, Y.G. and J.P.; project administration, Y.G. and J.P.; funding acquisition, Q.L. All authors have read and agreed to the published version of the manuscript.

**Funding:** This work was supported in part by the National Natural Science Foundation of China under Grant 62201209 and Natural Science Fund of Hunan Province under Grant 2022JJ40092.

**Data Availability Statement:** All data that support the findings of this study are available from the corresponding author upon reasonable request.

**Acknowledgments:** We gratefully thank the three anonymous reviewers for their valuable comments.

**Conflicts of Interest:** The authors declare no conflict of interest.

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
