# Peer review of "Highly Efficient Anchor-Free Oriented Small Object Detection for Remote Sensing Images via Periodic Pseudo-Domain"

_remotesensing, doi:10.3390/rs15153854_

Round 1

Reviewer 1 Report

This study investigates a high efficient anchor-free oriented small objects detection methods for remote sensing images. The topic fits well with the journal and special issue. Before the final recommendation, the reviewer has the following comments for further consideration by the authors:

- Methodology:

(1) Aiming at the problem that the rotation angle of high aspect ratio and low aspect ratio objects has different effects on regression, this paper designs the corresponding network structure and loss function. The insight of this paper is very meaningful, while the experimental part is relatively weak, which did not better highlight the advantages of the proposed method for high aspect ratio and low aspect ratio object detection. More sufficient experiments are recommended using more datasets.

(2) The authors mentioned that the experimental results found that an increase in the complexity of the model backbone leads to a decrease in the model’s performance and speed. This phenomenon is very interesting, but the author's explanation is relatively weak and not convincing enough. It is suggested that the authors: (a) replace the backbone of other methods of comparison with resnet-18, resnet-50, resnet-101, swin-t, and observe the change of the results; (b) use larger datasets, such as DOTA, to observe the network's ability to fit data; (c) use more efficiency indicators to characterize different methods, such as #Params, #FLOPs.

- Literature Review:

(1) Please enhance the literature review in depth. Related alghorithms and applications about oriented object detection and small dense object detection have been performed in the computer vision community, which include but are not limited to: https://doi.org/10.1016/j.engappai.2023.106686; https://doi.org/10.1111/mice.12940; https://doi.org/10.1109/TGRS.2020.3011418; https://doi.org/10.1016/j.autcon.2020.103124; https://doi.org/10.1177/147592172110042; https://doi.org/10.3390/rs14041012

(2) Ref. 4 and Ref. 10 are repetitive.

- Figures and Tables:

(1) It is recommended to draw all parts of the loss in the figure 5 at the same time.

(2) Figure 7 and Figure 8: font size should be modified to be consistent with the main text.

(3) The error bar or dispersion evaluation of the recognition metrics is recommended to add in the corresponding figures or tables.

Based on the above comments, the reviewer would recommend a major revision to further improve this manuscript.

Author Response

We sincerely appreciate the reviewer for his/her encouraging and positive comments.

- Methodology 

(1) Thanks for your appreciation of the proposed method. The experiment on a larger dataset will be more persuasive on the method. On the other hand, the motivation of this paper is to propose a simple way to tackle significant loss increases in boundary conditions. Larger datasets are mainly faced with multi-scale features, complex backgrounds, and other difficulties which may cause a shift in our center motivation. So it’s a pity we didn’t use a larger dataset. We are really sorry about that.

(2) Thanks a lot for pointing this out. Since the fact is not convincing enough, we revised words to “may leads to a decrease”.
  (R_a) Thanks for your valuable perspective. We searched for relevant literature using different backbones, which provided different results. Thus, the dataset size is not the only thing that decides which backbone is more proper. More details are provided in R_b.
  (R_b) On a larger dataset such as DOTA, the model with backbone R101 gets more mAP than the backbone with R50(Report by CFC-Net, Figure 6). And in the work “Dynamic Anchor Learning,” the model with backbone R101 also gets more mAP than backbone with R50 on a smaller dataset HRSC2016. This means the size of the dataset is not the only factor for the backbone. Even in a smaller dataset, a complex method may complement with complex backbone. The reason why our method performs better in a simple backbone might be the straightforward regression structure that we used. That is indeed an interesting open question.
  (R_c) As suggested by the reviewer, we add #Params and #FLOPs in Tab2.

- Literature

(1) Many thanks for the valuable literatures. We add all above algorithms to our reference except https://doi.org/10.1177/147592172110042 which can not be found in doi.org. And we enrich the section 2.2 and add more references. All added text has been labeled in red color.

(2) We sincerely thank the reviewer for careful reading. We are really sorry for our careless mistake. We have deleted a repetitive Ref in the revised manuscript.

- Figures and Tables:

(1) Many thanks for the valuable advice. As suggested by the reviewer, we draw all 5 parts of the loss in the figure 5.

(2) Sorry for the wrong font size in figure 7 and figure 8. As suggested by the reviewer, we modified the font size in figure 7 and figure 8 to the proper size.

(3) Many thanks for the advice. As suggested by the reviewer, we added the error bar in the revised manuscript as Figure 12.

If there are any other modifications we could make, we would like very much to modify them and we really appreciate your help.

Reviewer 2 Report

1)      The importance of the design carried out in this manuscript can be explained better than other important studies published in this field. I recommend the authors to review other recently developed works.

2)      "Discussion" section should be edited in a more highlighting, argumentative way. The author should analysis the reason why the tested results is achieved.

3)      It will be helpful to the readers if some discussions about insight of the main results are added as Remarks.

4) Proposed method should be detailed. Which was the method developed by authors?

5)   Similarly, "Conclusion" section should be rearranged. Taking advantage of these results, striking suggestions can be made for future studies.

6) What makes the proposed method suitable for this unique task? What new development to the proposed method have the authors added (compared to the existing approaches)? These points should be clarified.

This study may be consider for publication if it is addressed in the specified problems.

Author Response

We greatly appreciate the reviewer for his/her valuable comments and suggestions.

1) Many thanks for the valuable advice. As suggested by the reviewer, we enrich the section 2.2, and add more references in the revised manuscript. All added text has been labeled in red color.

2) Thanks a lot for pointing this out.  As suggested by the reviewer, we added the analysis of the reason at start of the “Discussion” section. All added text has been labeled in red color.

3) Many thanks for the advice. As suggested by the reviewer, we add the discussion about insight of main result in the “Discussion” section at second last paragraph. All added text has been labeled in red color.

4) Sorry for not being clear enough on this point. As suggested by the reviewer, we add more detail of proposed method at start of the “Materials and Methods” section. All added text has been labeled in red color.

5) Many thanks for the valuable advice. As suggested by the reviewer, we enrich the “Conclusion” section with suggestions and future studies. All added text has been labeled in red color.

6) Thanks for your suggestion. As reply in 4), we add more detail in the “Materials and Methods” section and clarify the new development to the proposed method also in “Discussion” section. All added text has been labeled in red color.

If there are any other modifications we could make, we would like very much to modify them and we really appreciate your help.

Reviewer 3 Report

The problems with the paper are as follows:

1.The text in Figure 7 is too large, it is recommended to adjust it smaller.

2. The writing of part 2.2 is too simple, it is recommended to add an introduction and analysis of relevant literature.

3.It is recommended to add a comparison of YOLO series in Table 3 to demonstrate the advantages and effectiveness of our algorithm.

4.The conclusion needs to further highlight the contribution of this article and provide prospects.

The workload of this article is average, and the level of innovation is slightly insufficient.

Minor editing of English language required.

Author Response

We sincerely appreciate the reviewer for his/her encouraging and positive comments.

1. Sorry for the wrong font size in figure 7. As suggested by the reviewer, we modified the font size in figure 7 to the proper size.

2. Many thanks for the valuable comment. As suggested by the reviewer, we enriched part 2.2 and added more references in the revised manuscript. All added text has been labeled in red color.

3. Thanks for your suggestion. As suggested by the reviewer, we added performance on the darknet to Table 3 in the revised manuscript.

4. Many thanks for the valuable advice. As suggested by the reviewer, we enrich the “Conclusion” section with suggestions and prospects. All added text has been labeled in red color. 

If there are any other modifications we could make, we would like very much to modify them and we really appreciate your help.

Round 2

Reviewer 1 Report

I have checked the revised manuscript. The previous concerns have been addressed, and no further comments are provided. The final acceptance is recommended.

Reviewer 2 Report

This paper could be accepted for publication

This paper could be accepted for publication

Reviewer 3 Report

Accept in present form

Accept in present form